# Generation of ultrahigh-brightness pre-bunched beams from a plasma cathode for X-ray free-electron lasers

Xinlu Xu [1✉], Fei Li[2], Frank S. Tsung[3], Kyle Miller [2], Vitaly Yakimenko[1], Mark J. Hogan [1], Chan Joshi[2] & Warren B. Mori[2,3]

The longitudinal coherence of X-ray free-electron lasers (XFELs) in the self-amplified spontaneous emission regime could be substantially improved if the high brightness electron beam could be pre-bunched on the radiated wavelength-scale. Here, we show that it is indeed possible to realize such current modulated electron beam at angstrom scale by exciting a nonlinear wake across a periodically modulated plasma-density downramp/plasma cathode. The density modulation turns on and off the injection of electrons in the wake while downramp provides a unique longitudinal mapping between the electrons' initial injection positions and their final trapped positions inside the wake. The combined use of a downramp and periodic modulation of micrometers is shown to be able to produces a train of high peak current (17 kA) electron bunches with a modulation wavelength of 10's of angstroms - orders of magnitude shorter than the plasma density modulation. The peak brightness of the nano-bunched beam can be $O(10^{21}A/m^2/rad^2)$ orders of magnitude higher than current XFEL beams. Such prebunched, high brightness electron beams hold the promise for compact and lower cost XEFLs that can produce nanometer radiation with hundreds of GW power in a 10$s$ of centimeter long undulator.

[1] SLAC National Accelerator Laboratory, Menlo Park, CA, USA. [2] Department of Electrical Engineering, University of California Los Angeles, Los Angeles, CA, USA. [3] Department of Physics and Astronomy, University of California Los Angeles, Los Angeles, CA, USA. ✉email: xuxinlu@slac.stanford.edu

X-rays with wavelengths ranging from ~100 to ~1 Angstroms have made great contributions in modern science, industry and medicine. X-ray free-electron-lasers (XFELs) can deliver directional and coherent X-rays at tunable wavelengths[1,2]. The XFEL process relies on an instability that arises when a bright electron beam propagates in an undulator that wiggles the electrons transversely. Current XFELs that use electron beams from a conventional accelerator operate in the self-amplified spontaneous emission (SASE) mode. Recently, SASE-FEL action in the XUV (27 nm) has been demonstrated by using electrons from a laser-plasma accelerator[3]. Unfortunately, SASE action produces photons with a limited longitudinal coherence. However, prebunching of the electron beam before injection into the FEL interaction region would produce coherent radiation emission at high harmonics of the bunching frequency, enhance the exponential gain of the FEL radiation power, and also enable high power and high radiative energy extraction efficiency by tapering-enhanced superradiance[4]. A coherent seed laser pulse co-propagating with the electrons is usually used to prebunch the beam. Due to the lack of useful seeds at X-ray wavelengths, harmonic seeding schemes, such as cascaded high-gain harmonic generation (HGHG)[5–9] and echo-enabled harmonic generation (EEHG)[10–16], have been studied extensively. In these schemes, one or two laser pulses, multiple undulators and magnetic chicanes are used to convert the wavelength of the electron density modulation from the seed laser wavelength ($\hbar\omega \sim 3$ eV) to its high harmonics with a harmonic number $h \leq 101$ ($\hbar\omega \sim 300$ eV)[9,16].

Due to its ability to sustain GV/cm acceleration gradient, plasma-based acceleration (PBA) driven by either an intense laser pulse or a high current charged particle beam[17–20] can accelerate electrons to GeV-level energies in only a few centimeters[21–24]. Furthermore, numerical experiments show that high quality (i.e., high brightness, low energy spread) beams suitable for driving an XFEL in the SASE mode[3] can be generated from PBA[25–27] thereby potentially reducing the size and cost of such machines. Beams with separation at the period of the plasma wave have been produced[28] which can radiate coherently at terahertz frequencies. Recent work has shown how laser-triggered ionization injection may be used to produce beams with density modulations at ~100 nm ($h \leq 5$)[29,30] and the generation of a single bunch with ~100 nm length by using a density bump[31]. To date no feasible way has been proposed to produce beams with angstrom density modulation in PBA.

In this article, we report on how to generate a GeV-level high-quality electron beam directly from PBA whose density is modulated in the X-ray wavelength range and with a harmonic number as high as $O(1000)$. The beam is produced from self-injection in a plasma wave wake created by a laser or particle beam driver as the driver transits a modulated plasma-density downramp (see Fig. 1). The plasma-density modulation in the downramp is driven by separate lasers. To facilitate comparison with previous conventional harmonic seeding schemes that use a laser[5,10], here the harmonic number is defined as the ratio of the wavelength of these lasers and the wavelength of the modulated electron beam. The ultrahigh acceleration gradient of the plasma wake makes the proposed scheme more compact and lower-cost than conventional seeded XFELs. Ultra-bright prebunched beams could produce fully coherent X-rays with hundreds of GW stable power and femtosecond duration by going through a 10s of centimeter long undulator. Such super-compact high power fully coherent X-ray sources could enable novel applications and generate great general interest in fields as diverse as high-density-density physics and atomic, molecular, and optical physics. The proposed scheme can also be a favorable complement to XFEL facilities by using their beams as drivers to improve the brightness of the bunched beams by orders of magnitude.

## Results

**Controlling injection through density modulation.** We propose to generate a microbunched electron beam by triggering a series of periodic downramp injections by sinusoidally modulating the density in a plasma downramp. When a short and intense driver propagates in a underdense plasma with speed $v_d \approx c$, a fully blowout plasma wave wake can be created behind it, where $c$ is the speed of light in vacuum. The phase velocity of this wake in a density gradient $n_p(z)$ can deviate from the driver's velocity[32] due to the density dependence of the oscillation frequency[33–37] as $v_\phi = \frac{v_d}{1 - (d\omega_p/dz)\omega_p^{-1}\xi}$, where $\omega_p(z) = \sqrt{\frac{n_p(z)e^2}{m_e\epsilon_0}}$ is the local plasma frequency, $\xi \equiv ct - z$, $m_e$ and $e$ are the electron mass and charge. A plasma with a density downramp region has been proposed to decrease the wake phase velocity and trap plasma electrons[33–37]. The injection condition is approximately described as $\gamma - v_\phi \frac{p_z}{m_e c^2} = 1 + \frac{e}{m_e c^2}\psi$, where $\gamma, p_z$ are the relativistic factor and axial momentum of the electron when it is injected (i.e., $v_z = v_\phi$) and $\psi \equiv \phi - \frac{v_\phi}{c}A_z$ is the wake potential. It shows that for a given minimum of $\psi$ there is a threshold for $v_\phi$ for which self-injection occurs, which is $v_{\phi,\text{th}} \approx 0.9998c$ for the parameters considered here.

The phase velocity of the wake can be modulated when a driver propagates through a plasma with a density modulation as $\delta\bar{n}\sin(k_m z)$[31], where $\delta\bar{n}$ and $k_m$ are the amplitude and wavenumber of the modulation. Substituting the expression of the density into the phase velocity expression gives $\frac{\delta v_\phi}{c} \approx \frac{k_m}{k_{p0}}\frac{\delta\bar{n}}{2n_{p0}}k_{p0}\xi\cos(k_m z)$, where $k_{p0} = \frac{\omega_{p0}}{c}$. Thus the phase velocity can oscillate between subluminal and superluminal so that injection can be turned on ($v_\phi < v_{\phi,\text{th}} \approx c$) and off ($v_\phi > v_{\phi,\text{th}} \approx c$) periodically. However, a micrometer scale density modulation of the plasma, which is much shorter than the plasma wake wavelength $\lambda_w$ (10s of microns), is needed to achieve a nanoscale modulation of the injected beam. This is drastically different than considered in previous work where $\frac{2\pi}{k_m} \gg \lambda_w$[20,25,31,33–38]. The electrons experience a density varying ion background during their oscillations. This violates the assumption behind the phase velocity expression that the electrons oscillates around ions with locally constant density. We find the phase velocity is still periodically modulated at the wavenumber $k_m$ in this case, although with a reduced amplitude (see Supplementary Note 1).

Such a density modulation provides discrete injection by turning the injection on and off. However, a downramp is needed to map the injection at each period to different $\xi$ locations to form a density modulated beam. After injection, the downramp leads to electrons being mapped to the tail of the wake sequentially, i.e., the electrons injected earlier sit in front of the electrons injected later (Fig. 1a) due to the gradual elongation of the bubble in the downramp. The net result is a macro-bunch with periodic micro-bunches. The quality of the macro-bunch has similar properties as a beam from normal density downramp injection[25], i.e., it has high current, low emittance and low energy spread; thus it may be used to produce fully coherent, stable, hundreds of gigawatt X-rays with femtosecond duration in a short resonant undulator.

A premodulated plasma (both ions and electrons) is considered above. A technique for generating such ion density gratings in plasmas has recently been demonstrated[39]. In the ab initio simulations presented here, we use an alternative method to produce the plasma-density modulation that utilizes the ponderomotive force of two counter-propagating lasers (see Methods section)[40]. In this case, the plasma electrons are modulated with a wavenumber $k_m = 2k_L$. However, the ions are not modulated because the ponderomotive force on the ions is negligible and the space-charge force due to the electron modulation does not affect

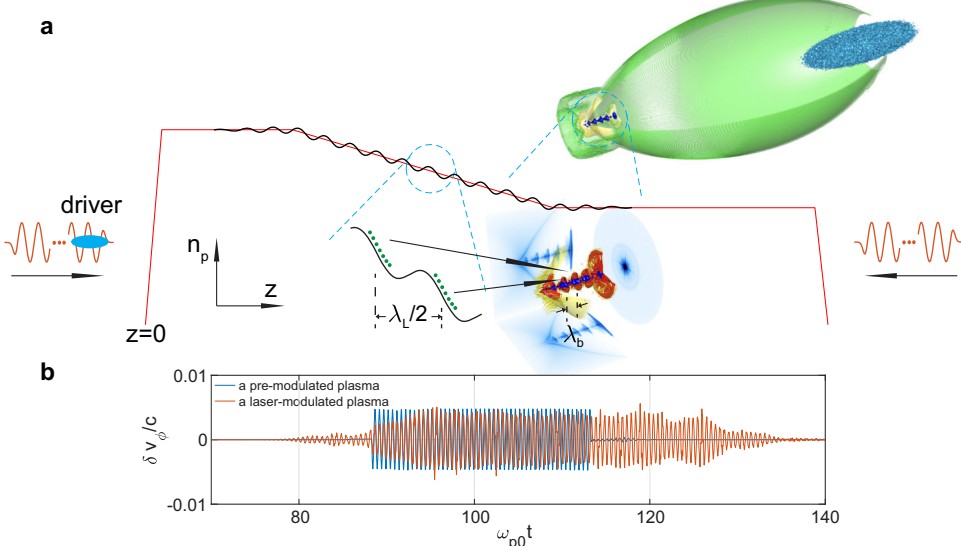

**Fig. 1 The concept of bunched beam generation from density downramp injection in PBA (not to scale). a** A plasma with a density downramp (red line) is set at $t = 0$. Two linearly polarized laser pulses propagate from the two ends of the plasma towards its center and their overlap creates a density modulation with period $\lambda_L/2$ (black line). A properly delayed electron beam driver is incident on the plasma from the left to excite a blowout wake (as shown in the upper inset) and inject electrons only at half of the density modulation period (green dots) and they are mapped to discrete axial positions with period $\lambda_b$ after injection (as shown in the lower inset). **b** The modulation of the phase velocity of position $E_z = 0$ with $k_{p0}\xi \approx 4$.

the ions during the duration of the short laser pulses considered here. We find that the modulation of the ponderomotive force acts similarly to the charge separation force of the modulated ion density in a premodulated, i.e., ion density grating, case (see Supplementary Note 1). The modulation of the phase velocity from a premodulated plasma and a laser-modulated plasma are shown in Fig. 1b. The phase velocities are modulated with $k_m = 2k_L(= 4\pi k_{p0})$ and an amplitude $0.005c$ which is much smaller than the predication ($\delta v_\phi \approx 0.05c$ when $\delta\bar{n} = 0.002n_{p0}$ and $k_{p0}\xi \approx 4$). Here $k_L$ was set to $2\pi k_{p0}$ for convenience.

**Bunched beam generation.** To demonstrate the bunched beam generation, we use the quasi-3D version[41] of the fully relativistic PIC code OSIRIS[42] with a recently developed customized Maxwell solver which can model relativistic beam propagation with high fidelity[43]. As illustrated in Fig. 1a, in ab initio simulations a plasma starts from $z = 0$ with a density $1.1n_{p0}$ that drops linearly to $n_{p0}$ from $37.5k_{p0}^{-1}$ to $62.5k_{p0}^{-1}$ $\left(g \equiv \frac{1}{n_p}\frac{dn_p}{dk_p z} \approx 0.004\right)$ and extends to $100k_{p0}^{-1}$ with $n_{p0}$. Two laser pulses, polarized along $\hat{x}$ with $a_{L0} = 0.005$ and $\omega_L = 2\pi\omega_{p0}$ are sent from both ends of the plasma and propagate toward one another. The lasers have a rising and falling edge $10\omega_{p0}^{-1}$ and a plateau $70\omega_{p0}^{-1}$, and a Gaussian transverse profile with a spot size of $6k_{p0}^{-1}$ at focus and a focal plane of $50k_{p0}^{-1}$. A plasma-density modulation with wavelength $\lambda_m = \frac{\lambda_L}{2}$ is observed around the ramp region as shown in Fig. 2a. The black line shows the modulation amplitude at $45.5k_{p0}^{-1}$ which varies along the transverse directions due to the Gaussian transverse distribution of the lasers. The blowout radius of the wake is $\sim 4k_{p0}^{-1}$, so the modulations are relatively constant in the region of physical interest.

The electron beam driver has an energy of $E_d = 2\,\text{GeV}$, a peak current of $I_d = 34\,\text{kA}$, a spot size of $k_{p0}\sigma_r = 0.5$, a duration of $k_{p0}\sigma_z = 0.7$, and a center of $k_{p0}z_c = -47.1$ at the beginning of the simulation. Plasma electrons are completely expelled by the electric field of the driver and some of them are pulled back toward the axis by the immobile ions and form a high-density sheath[44]. When propagating through a region of decreasing density, the wake

expands, and some sheath electrons are injected at the tail end of the wake. These electrons are longitudinally locked with the driver since they are both highly relativistic. We track these injected electrons and show their initial positions $z_i$ and their relative positions inside the wake $\xi$ after injection in Fig. 2b. Thus, the position $\xi$ for a particle depends on its initial location in the ramp $z_i$. It is obvious that the injection position is modulated at the plasma-density modulation wavelength ($0.5k_{p0}^{-1}$) as shown in the enlarged inset. There is a mapping between the initial and final position of an electron after injection[25], i.e., $\frac{d\xi(z_i)}{dz_i} \approx 4.5g$ where the wake wavelength $k_p(z)\lambda_w \approx 9$ observed from simulations is used. Ramps with nonlinear profiles can introduce a chirp of the bunching wavenumber along the beam. Clearly, the macro length and micro modulation period are compressed by a factor of $\frac{1}{4.5g} \approx 55.6$ for $g = 0.004$. The harmonic number is $h = 2\left(\frac{d\xi}{dz_i}\right)^{-1} \approx 0.44g^{-1}$, which is $h \approx 111$ for $g = 0.004$. We note there is also some discrete injection before the ramp and near the end of the ramp, however the electrons injected from these regions are not bunched, as evidenced by constant value of $\xi$ even though $z$ is changing, due to the lack of the mapping which is only present in the ramp. This clearly demonstrates that both the downramp and the periodic modulation are required to provide prebunched beams.

The density distribution of the injected electrons is shown in Fig. 2c where the bunched structure can be seen. A bunching factor $b(k) \equiv |\sum_{j=1}^{N}\exp(ikz_j)|/N$ is usually used to quantify the modulation, where $N$ is the total number of the electrons. As shown in Fig. 2d, the bunching factor of the injected beam reaches its maximum at $k \approx 105k_L$ with $b \approx 0.042$. The bunching factor of the injected electrons when a premodulated plasma is used is also shown which is similar to the one from the ponderomotive force modulated plasma. We note that only electrons with $k_{p0}\xi \geq 9.85$ (or equivalently $-k_{p0}\xi \leq -9.85$) are used to calculate the bunching factor.

The longitudinal phase space of the injected electrons and its current profile at $t = 145\omega_{p0}^{-1}$ are shown in Fig. 2e, where the bunched structure is seen clearly. The beam has a positive energy

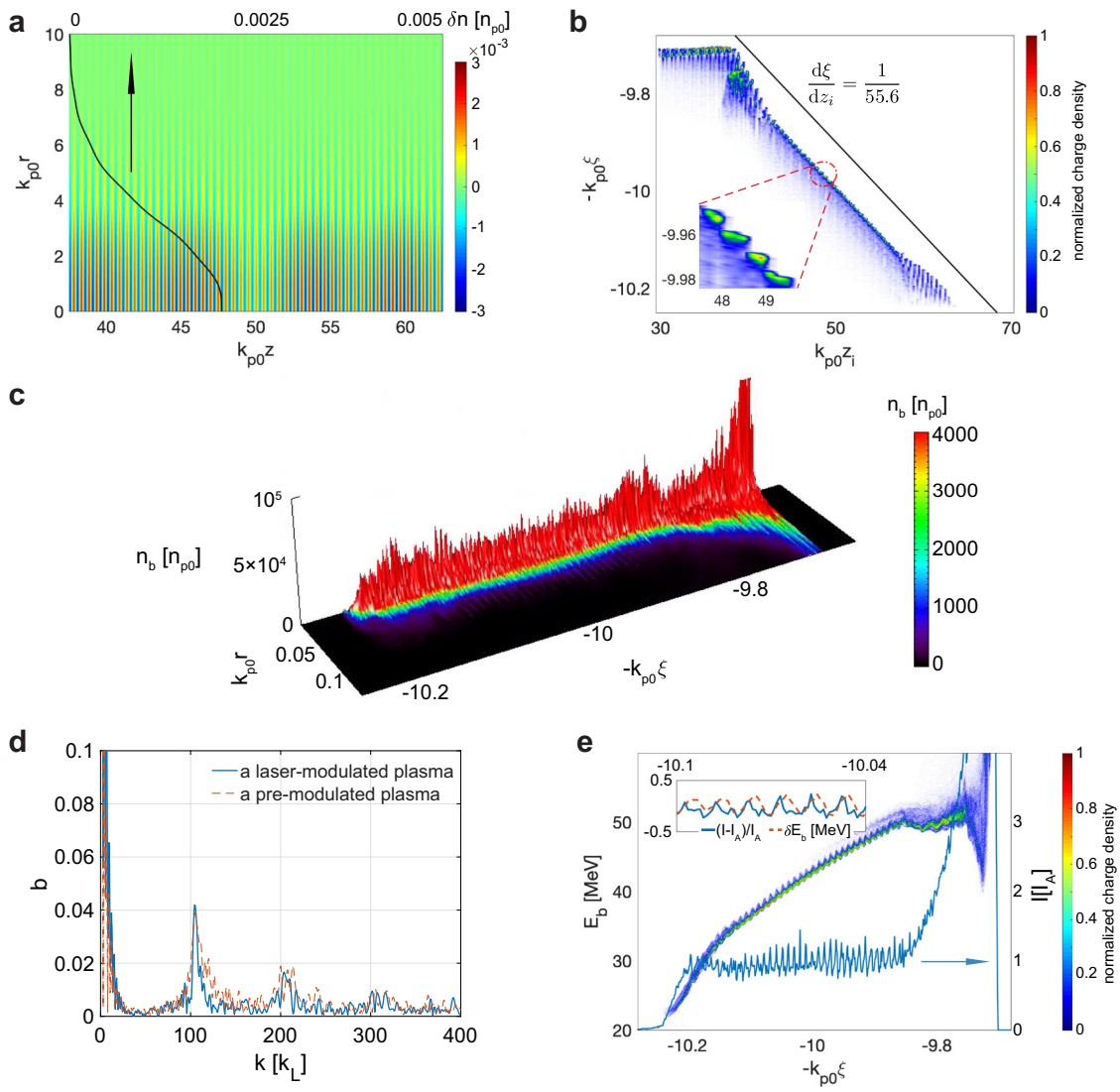

**Fig. 2 Generation of bunched electrons from quasi-3D PIC simulations. a** The perturbed plasma density of the ramp region in the $r-z$ plane at $100\omega_{p0}^{-1}$. The black line is along $k_{p0}z=45.5$. **b** The relation between the initial longitudinal positions of the injected electrons and their positions inside the wake. **c** The density distribution of the injected electrons at $145\omega_{p0}^{-1}$. **d** The bunching factor at $145\omega_{p0}^{-1}$. **e** The longitudinal phase space of the injected electrons and their current profile (blue line) at $145\omega_{p0}^{-1}$. The inset shows the chirp-corrected energy modulation and the current.

chirp $\left(\frac{dE_b}{dz}>0\right)$ since the electrons at the head are injected at earlier times and are thus accelerated over a longer distance[25]. A sinusoidal energy modulation with an amplitude $\sim$ MeV is present. This is caused by the axial space-charge interactions between the bunched electrons while they are at low energies because the axial electric field of relativistically moving electrons decreases rapidly with their energy ($E_{SC,z}\propto\gamma^{-2}$). The chirp-corrected average energy modulation and the current profile of the beam are shown in the inset of Fig. 2e. We can see the energy modulation is approximately $\frac{\pi}{2}$ in phase ahead of the current (density) modulation which confirms that the space-charge force from the current modulation is the cause of the sinusoidal energy modulation. The degradation of the bunched structure induced by this energy modulation when the beam is boosted to higher energy is negligible. The slippage induced by the energy modulation is $\delta s \approx \int dz \frac{\delta\gamma}{(\gamma_i+eE_zz/m_ec)^3} \leq \frac{m_ec}{2eE_z}\frac{\delta\gamma}{\gamma_i^2} \approx 10^{-4}k_{p0}^{-1}$, which is much less than the bunching wavelength $0.01k_{p0}^{-1}$ when $E_z = 2\frac{m_ec\omega_{p0}}{e}$, $\delta\gamma = 2$, and $\gamma_i=80$ are substituted into the formula. The

degradation of the bunched structure due to the transverse betatron motion can also be shown to be negligible using similar arguments, $\delta s \approx \int dz \frac{(p_\perp/m_ec)^2}{2(\gamma_i+eE_zz/m_ec)^2} \leq \frac{m_ec}{2eE_z}\frac{(p_{\perp,i}/m_ec)^2}{\gamma_i^{1/2}} \approx 10^{-4}k_{p0}^{-1}$, where $p_{\perp,i} = 0.04m_ec$ is the full width of the maximum of the transverse momentum spread at $145\omega_{p0}^{-1}$. We accelerate the beam to 1.08 GeV numerically in the Supplementary Note 3 and show the bunching factor changes little.

The bunched beam is characterized by a slice energy spread of $\sim$0.4 MeV, an average current of $\sim$17 kA, and a normalized emittance of $\sim 0.006k_{p0}^{-1}$, which is suitable to drive an XFEL. There is also a large energy chirp [$\sim$50 MeV/($k_{p0}\xi$)] formed during the injection process. However, this chirp can be mitigated by the inverse chirp naturally imposed by the acceleration gradient inside the wake[44] (electrons injected first reside farther forward in the wake where the acceleration gradient is smaller). Thus, there is an optimized acceleration distance where the beam achieves a flatter longitudinal phase space[25] that minimizes the projected energy spread of the macro-bunch.

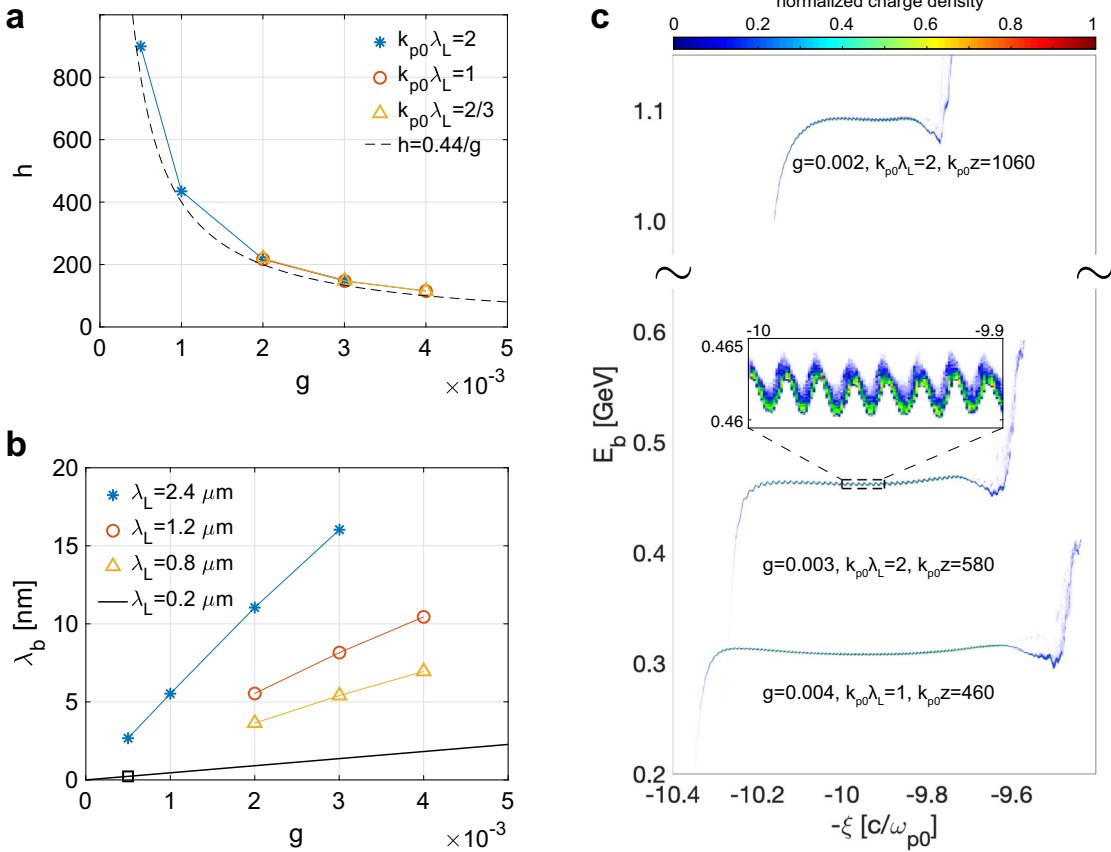

**Fig. 3 Tunability of the bunching wavelength and flat longitudinal phase space of the injected beams. a** The dependence of harmonic number $h$ on the normalized density gradient $g$ under different colliding lasers. **b** The modulation wavelength when different ramps and lasers are used where $n_{p0} = 1.97 \times 10^{19}$ cm$^{-3}$. **c** The longitudinal phase space of the injected electrons at their optimized acceleration distance. The inset shows the sinusoidal energy modulation.

**Tunability of the bunching wavelength and flat longitudinal phase space.** In order to understand how to choose plasmas parameters to optimize the process, we performed a wide parameter scan using quasi-3D OSIRIS with only the $m = 0$ mode where a density modulation with $\delta\bar{n} = 10^{-3}n_{p0}$ is initialized on top of a plasma downramp. A moving window is used since we are not self-consistently including lasers to create the modulation (see Methods section). Due to the reduction in computational needs, we can also simulate the acceleration of the beam to ~GeV energies. To isolate the physics, we also employ a non-evolving beam driver to model the injection and acceleration. The dependence of the harmonic number $h$ on the gradient of the ramp $g$ is shown in Fig. 3a (where the theory curve $h = \frac{0.44}{g}$ is also plotted). When using a ramp with $g = 5 \times 10^{-4}$ ($\Delta n = 0.05 n_{p0}, k_{p0}L = 100$), the modulation frequency of the injected electrons is as high as 450 times the modulation frequency or 900 times of the laser frequency, i.e., $h \approx 900$, which is one order of magnitude higher than the number from staged HGHG ($h \leq 60$[9]) and EEHG ($h \leq 101$[16]). Normalized units are used in the above results where the modulation frequency is scaled to the plasma frequency. We can choose a normalized density to make connections with possible near-term experiments. Figure 3b shows the bunching wavelength vs. $g$ for various $\lambda_L$ for $n_{p0} = 1.97 \times 10^{19}$ cm$^{-3}$ (assuming that the density modulation wavelength is $\lambda_L/2$) and it can be seen that modulations at several nm are achieved. In principle, beams with shorter modulated wavelength can be injected if the ramp density gradient is smaller or the density modulation wavelength of the ramp is shorter, which might be the case for shorter wavelength lasers. The predicted results when $\lambda_L = 0.2$ μm is

shown in black line in Fig. 3b where 0.23 nm bunching wavelength is achieved when $g = 5 \times 10^{-4}$.

In Fig. 3c, we show that when the injected electrons are accelerated to an optimized distance[25] the projected energy spread of the beam core is <1 MeV. Since the initial energy chirp induced during the injection process is inversely proportional to the gradient of the ramp, i.e., $\frac{dE_b}{dz} \propto g^{-1}$, a longer acceleration distance is required to remove the chirp when $g$ is small. Therefore, this will also lead to a higher average beam energy. Beam loading[45] whereby the beam itself modifies the accelerating field can also play a role during the removal of the chirp and this is self-consistently included in the simulations.

## Discussion
As noted above the longitudinal space charge can create a ~MeV sinusoidal energy modulation along the beam as shown in the inset of Fig. 3c. By letting the beam go through a chicane where the electrons' axial location changes according to their energy, this energy modulation could be utilized to enhance the bunching factor at the $hk_L$ and their harmonics as in a HGHG. For example, numerical calculations show a small chicane with $r_{56} = \frac{\Delta z}{\Delta E_b} = 4.1$ nm/MeV can increase the bunching factor at the fundamental frequency of the beam generated with $g = 0.003, k_{p0}\lambda_L = 2$ from 0.05 to 0.24, where $\Delta z$ is the displacement of an electron with a energy offset $\Delta E_b$ with respect to a reference electron with an energy $E_{b0} \approx 0.46$ GeV. Such energy modulation will allow experimentalists to measure the bunching structure by dispersing the electrons in a magnetic dipole.

In order to transport the beams out of the plasma into an undulator without emittance degradation a plasma matching section with gradually varying density is needed[46–48] to match the beam. The matched beam can then travel through a resonant undulator to generate fully coherent radiation. Due to the high current (~17 kA) and low emittance ($\sim 0.01 k_{p0}^{-1} \approx 12$ nm) of these beams, hundreds of GW of radiation can be emitted in a short beam-undulator interaction distance.

We consider the 1.09 GeV injected beam from $g = 0.002$, $\lambda_L = 0.8$ µm case as an example for a GENESIS simulation[49]. Figure 4 shows that 3.6 nm radiation grows exponentially along the undulator with an e-folding gain length $L_g \approx 4$ cm and that the output radiation saturates at $z \approx 0.3$ m with $P \approx 234$ GW (see Methods section and Supplementary Note 5). The planar undulator has a wavelength as $\lambda_U = 1$ cm and its normalized vector potential amplitude is $K \equiv \frac{eB_0\lambda_U}{2\pi m_e c} = 2.13$, where $B_0$ is the magnetic field on axis. The FEL interaction is in the cold and tenuous beam limit[50].

In order to excite the plasma wake effectively, the major requirements of the beam driver are that it has a high peak current and its size needs to be smaller than the plasma skin depth in each dimension. On the other hand, there are no stringent requirements for the energy, energy spread and emittance of the driver (see Supplementary Note 6). The ultra-short electron beams produced at the FACET-II facility[51] and current XFEL facilities can serve as drivers to produce bunched beams with orders of magnitude higher brightnesses. These existing beams might have spot sizes that are too large, but they can be focused down by properly designed plasma upramps[52,53]. However, in order to reduce the cost or make the process more compact, there are possible research paths to pursue. A driver with lower current or non-ideal size excites a smaller wake which would generate a bunched beam with larger harmonic number $h$ (see Supplementary Note 6) but a lower peak current[25,54]. The electron beams produced in laser-plasma wakefield accelerators are also suitable to be used as drivers[55]. Laser pulses could also be used directly as drivers to produced bunched beams (see Supplementary Note 6).

In conclusion, we propose using PWFA, and the combination of a density downramp and a density modulation to produce ultra-bright and high-quality electron bunches with a current modulation at X-ray wavelengths. We have presented fully self-consistent ab initio OSIRIS PIC simulations that show that the ponderomotive force of two counter-propagating lasers that overlap in the downramp can produce a sufficient density modulation with a wavelength half of the laser wavelength. The current of the injected beam is modulated at a wavelength $O(1000)$ times smaller than the modulation wavelength in the downramp.

## Methods

**Modulation of the plasma electron density using two colliding lasers**. Generating electron density modulations in a downramp can be realized through the ponderomotive force of two counter-propagating linearly polarized laser pulses of the same frequency[40]. The modulation will be at half the wavelength of lasers so injection will turn on and off with a length scale of half the optical wavelength. As illustrated in Fig. 1, two non-relativistic lasers ($a_L \equiv \frac{eA_L}{m_e c} \ll 1$ where $A_L$ is the vector potential of the laser) with identical frequencies propagate into the plasma from both ends and interfere inside the density ramp. This results in an electron density modulation (red line in Fig. 1) with an amplitude determined through the balance of the ponderomotive force on the electrons with the charge separation force between the electrons and ions.

The amplitude of the density modulation can be estimated from $\frac{\partial^2 \delta n}{\partial t^2} + \omega_p^2 \delta n = \frac{e^2 n_p}{2 m_e^2} \frac{\partial A_L^2}{\partial z^2}$ where $\delta n \ll n_p$ is assumed. If the envelop of the lasers rises slowly compared with the plasma oscillation, $\delta n$ follows the envelope of $A_L^2$. For the lasers with an approximately trapezoid shape, the perturbed density saturates at $\delta n/n_p \approx 2a_{L0}^2(\omega_L/\omega_p)^2 \cos(2k_L z)$[40] (see Supplementary Note 2). 1D particle-in-cell (PIC) simulations are conducted to confirm the creation of this density modulation. A cold plasma with constant density $n_{p0}$ is distributed between $z = 0$ and $100 k_{p0}^{-1}$ and two lasers pulses with $\omega_L = \pi\omega_{p0}$, $a_{L0} = 10^{-3}$ propagate towards one another from the two ends of the plasma. The plasma electron density perturbation is shown in Fig. 5a where a spatial sinusoidal density modulation is present in a rhombic region of the $z - t$ space. The inset shows the density seen by an observer moving at $c$ at a later time where the dashed line is its trajectory. The plasma density is modulated at wavenumber $k_m = 2k_L$ and its amplitude is $\delta\bar{n} \approx 2 \times 10^{-5} n_{p0}$. The frequency and intensity of the colliding lasers are scanned, and the amplitudes of the excited density modulation are summarized in Fig. 5b. Good agreement between the simulations and the formula are obtained. The ions do not respond to the ponderomotive force, so for the parameters considered here, the ions are essentially immobile.

**Production of Fig. 1b**. Two simulations are done for each case: a premodulated plasma case and a laser-modulated plasma case. For the premodulated case, the plasma density is uniform between 0 to $100 k_{p0}^{-1}$ in the first simulation and a

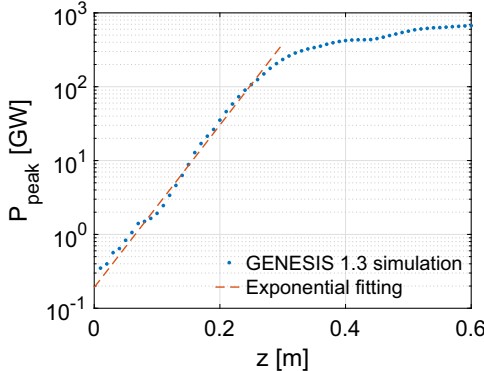

**Fig. 4 Simulation results of X-ray generation.** The peak power of the radiation along the undulator from GENESIS simulation (blue dots) and the exponential fitting (red line).

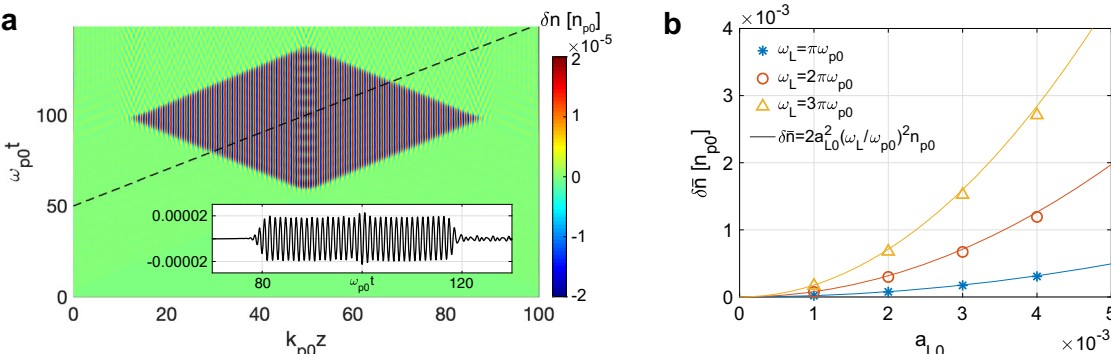

**Fig. 5 Laser-modulated plasma. a** The perturbed density distribution induced by the colliding lasers in the $z - t$ plane from 1D simulations. The inset shows the density along the dashed trajectory. The lasers have a rising and falling edge $10\omega_{p0}^{-1}$ and a plateau $60\omega_{p0}^{-1}$. **b** The dependence of the perturbed density on the frequency and intensity of the colliding lasers.

**Table 1 Parameters of the results in Fig. 3.**

| $k_{p0}\lambda_L$ | $\frac{\Delta n}{n_{p0}}$ | $k_{p0}L_r$ | $g$ | Grid numbers | Particles per cell |
|---|---|---|---|---|---|
| 2 | 0.15 | 50 | $3 \times 10^{-3}$ | $5760 \times 6144$ | $2 \times 2$ |
| 2 | 0.1 | 50 | $2 \times 10^{-3}$ | $5760 \times 6144$ | $2 \times 2$ |
| 2 | 0.1 | 100 | $1 \times 10^{-3}$ | $11520 \times 12288$ | $1 \times 2$ |
| 2 | 0.05 | 100 | $5 \times 10^{-4}$ | $23040 \times 12288$ | $1 \times 2$ |
| 1 | 0.2 | 50 | $4 \times 10^{-3}$ | $5760 \times 6144$ | $2 \times 2$ |
| 1 | 0.15 | 50 | $3 \times 10^{-3}$ | $11520 \times 6144$ | $2 \times 2$ |
| 1 | 0.1 | 50 | $2 \times 10^{-3}$ | $11520 \times 6144$ | $2 \times 2$ |
| 2/3 | 0.2 | 50 | $4 \times 10^{-3}$ | $11520 \times 6144$ | $2 \times 2$ |
| 2/3 | 0.15 | 50 | $3 \times 10^{-3}$ | $11520 \times 6144$ | $2 \times 4$ |
| 2/3 | 0.1 | 50 | $2 \times 10^{-3}$ | $18000 \times 6144$ | $2 \times 4$ |

Shape of the premodulated plasma downramp and the corresponding simulation parameters (grid numbers and particles per cell) in PIC simulations.

plasma-density modulation as $0.002n_{p0} \sin(4\pi k_{p0}z)$ is superimposed between $37.5k_{p0}^{-1}$ and $62.5k_{p0}^{-1}$ in the second simulation. The phase velocity variation $\delta v_\phi$ is defined as the difference between $v_{\phi,E_z=0}$ in these two simulations. For the case where the plasma electrons are modulated by two laser pulses, the two lasers have the same polarization which can modulate the electron in the first simulation and orthogonal polarization which cannot modulate the electron in the second simulation. The parameters of the electron beam driver and the laser pulses are as same as them in Fig. 2 in the main text. Note that the density modulation amplitude ($\delta \bar{n} = 0.002n_{p0}$) in the premodulated case is set as equal to the expected value generated by two laser pulses with the same polarization.

**Particle-in-cell simulations**. We use the quasi-3D version[41] of the fully relativistic PIC OSIRIS[42] with a recently developed customized Maxwell solver which can model the propagation of relativistic particles with high fidelity[43] to model the generation and acceleration of a bunched electron beam. The simulation shown in Fig. 2 is done using a fixed simulation box with dimensions of $280k_{p0}^{-1} \times 12k_{p0}^{-1}$ and with $143360 \times 3072$ grids along the $z$ and $r$ directions, respectively. This corresponds to a grid size along $z$ of $\frac{1}{512}k_{p0}^{-1}$ which is needed in order to resolve the short scales of the injected and trapped electrons. The time step is $dt = \frac{dz}{2c}$. Each cell contains 8 macro-particles to represent the beam driver and the plasma electrons respectively. Only the $m = 0$ and 1 modes (physical quantities are of the form $\exp(\pm i\phi)$, where $\phi$ is the azimuthal angle in the transverse plane) are included in order to model linearly polarized lasers. We have carried out simulations with 10 cells per bunching wavelength for steep density ramps with $g = 0.008$ to confirm the results for the fundamental bunching factor.

The simulation results shown in Fig. 3 for a premodulated downramp are done using only the $m = 0$ mode. A simulation box with dimensions $11.25k_{p0}^{-1} \times 12k_{p0}^{-1}$ moves along the $z$-direction with speed of light in vacuum to model the physics in the first wave bucket. See Table 1 for detailed parameters of the simulations. The time step is always set as $dt = \frac{dz}{2c}$. The high fidelity customized Maxwell solver developed in ref. [43] is used.

**GENESIS 1.3 simulation**. We use the 3D code GENESIS 1.3[49] to model the FEL interaction. The phase space distribution of the injected beam for the case $g = 0.002$, $\lambda_L = 0.8\,\mu m$ at $z = 1060k_{p0}^{-1}$ (1.27 mm) is output from the quasi-3D PIC simulation and then interpolated to six-dimensional phase space. Due to the ultrastrong focusing fields of the plasma wake, the injected beam has an extremely small spot size ($\sigma_r \approx 17$ nm) and $\beta$-function ($\beta = \sqrt{2\gamma_b}c/\omega_{p0} \approx 80\,\mu m$), where the beam energy is 1.09 GeV and the background plasma density is $n_{p0} = 1.97 \times 10^{19}$ cm$^{-3}$. We use a plasma matching section with profile $n_p(z) = \frac{n_{p0}}{\left[1 + (z - z_m)/l\right]^2}$ to enlarge the beam spot size, where $z_m$ is the start of the matching section. After a plasma matching section with $l \approx 47\,\mu m$ and $L = 0.1$ m, the injected beam has $\sigma_r \approx 1.4\,\mu m$, $\beta \approx 0.56$ m and $\alpha \approx -1.3$. The beam then drifts 10 cm in free space to reach the entrance of the undulator and has $\sigma_r \approx 1.8\,\mu m$, $\beta \approx 0.9$ m, $\alpha \approx -1.8$. The plasma matching section and the drift are modeled by the corresponding transfer matrix. The phase space distribution of the injected beam at the end of the drift is imported into GENESIS 1.3 with preserved current and energy modulation.

## Data availability
The data that support the findings of this study are available from the corresponding author upon reasonable request.

## Code availability
All computer codes supporting the findings of this study are fully documented within the paper and its references. Reasonable additional inquiries about the codes should be directed to the corresponding author.

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

## Acknowledgements

This work was supported by the U.S. Department of Energy under Contracts No. DE-AC02-76SF00515 and No. DE-SC0010064, the U.S. National Science Foundation under Grants No. 2108970, and the DOE Scientific Discovery through Advanced Computing (SciDAC) program through a Fermi National Accelerator Laboratory (FNAL) sub-contract No. 644405. The simulations were performed on the resources of the National Energy Research Scientific Computing Center (NERSC), a U.S. Department of Energy Office of Science User Facility located at Lawrence Berkeley National Laboratory, through an ALCC grant.

## Author contributions

X.X., F.L., F.T., K.M., V.Y., M.H., C.J., and W.M. contributed extensively to the work presented in this paper.

## Competing interests

The authors declare no competing interests.
