## [Peer Review File · Nature Communications]

Generation of ultrahigh-brightness pre-bunched beams from a plasma cathode for X-ray free-electron lasersREVIEWER COMMENTS

Reviewer #1 (Remarks to the Author):

This manuscript describes an original scheme for bunching the density of a plasma wakefield accelerated electron beam at high harmonic frequencies up to the X-Ray regime. It is proposed that injecting such a beam into an undulator would start an FEL gain process without seed radiation injection or without relying on the wide band spectrum of electron beam shot-noise in SASE FEL. Such a coherent radiation emission process is essentially a bunched-beam coherent superradiance process in the high-gain regime. Pre-bunching of an electron beam can be generated with laser beams by the processes of HGHG and EEHG after the beam is accelerated to high energies. However, the proposed concept here is radically different, and is specific to the scheme of plasma wakefield acceleration, in which case the density modulation of the electrons is done by spatial density modulation of stationary plasma with two counter propagating laser beams. The bunched charge is then self-injected and accelerated by the plasma wake field of a short high energy electron beam pulse

This concept looks like a promising way to utilize the relatively compact high acceleration gradient of PBA for operating an X-Ray FEL. The prebunching of the beam would enhance the FEL coherence and will expediate the FEL radiative power generation in a short undulator section (1 meter). The paper is in my mind of high value, and should be of interest to the scientific community of Science Communications readers. However, in my opinion it is not written in a clear and comprehensible way in its present form. Also, while the explanation of the bunching and plasma wakefield acceleration is detailed, the calculation of the gain and power of the pre-bunched beam FEL was done very briefly, possibly with not well fitting expressions that were derived earlier for a SASE FEL in another paper that is referred to. The FEL gain regime parameters are not given and the high power prediction is not well justified in the text or in a supplementary. I listed specific comments in an attachment, and I hope they will help the authors to improve the manuscript. I favor publication of this paper after substantial revisions.

Reviewer: Avraham Gover

Dec. 10, 2021

Review of a manuscript submitted to Nature Communications:

"Generation of ultrahigh-brightness, pre-bunched beams from a plasma cathode for X-ray free-electron lasers"

I will present my comments in two parts: comments regarding the estimate of X-Ray power generation with the proposed beam bunching scheme and comments regarding the clarity of presentation of the bunching scheme.

1. comments regarding the estimate of X-Ray power generation with the proposed beam bunching scheme

The authors proposed a scheme of generating a periodically "pre-bunched electron beam at Angstrom scale by exciting a nonlinear wakefield across a periodically modulated plasma-density downramp/plasma cathode". They assert that upon injection of the modulated beam into a short undulator, high order harmonic components of the modulated beam current will excite coherent high power FEL radiation (for example 0.3 TW at 3.6 nm with a 1.09 GeV e-beam). This estimate is justified hastily by relying on the SASE FEL formulas of Xie [51].

The idea of prebunching the electron beam with high harmonic contents at X-Ray frequency is very interesting and may be important. While the main focus of the paper is the acceleration and bunching process, I expect that the estimate of the impressive high power predicted by the authors would be justified in more details and rigor, at least in the supplementary:

-Which formula in Xie's paper is used, the exponential linear gain formula? The saturation power formula?

-The high gain formula in Xie's paper refers specifically to SASE FEL, where the input power is the beam shot-noise (spontaneous emission). In the present case the input is coherent current pre-bunching. What is the analytical formula used? Short of simulation of the FEL problem (that is probably out of the scope of the present loaded paper) one can use an analytic solution of the cubic equation with initial conditions (in particular, initial condition of current density bunching). I can suggest the paper of [Schnitzer] with which I am familiar, but there are probably more recent references.

-besides current modulation, the solution of the cubic equation can be given also in terms of velocity (energy) modulation. This may be relevant to the example in the paper where energy modulation is predicted (line 209). This energy modulation seems to be significant, since it can enhance the density bunching parameter from 0.05 to 0.24 with a short chicane (line 308). Since the wiggler is dispersive too, the energy modulation may play a role in the FEL dynamics in a meter long undulator.

-Since the beam parameters in the proposed scheme are a unique new case, it is desirable to check the operating parameters of the FEL dispersion equation for this case and verify the gain regime of the FEL (see [Gover and Sprangle] for example). Is the interaction in the cold beam limit? Is the exponential gain in the tenuous beam limit or there is a modification due to collective interaction? What is the gain e-folding length?

- The description of the contribution of reference [4] in the paper in lines 14-18 is not accurate. A more correct description would be in my words: Prebunching of the electron beam before

injection into the FEL interaction region can produce coherent radiation emission at high harmonics of the bunching frequency, enhance the exponential gain of the FEL radiation power, and also can enable high power high radiative energy extraction efficiency by tapering-enhanced superradiance (TES) [4].

- Incidentally, I suggest the authors to examine the scheme of self-interaction TES of Ref. 4, where the bunched beam is trapped by its own superradiant radiation emission in a tapered undulator. Quite high trapping efficiency and high-power extraction may be attained in this case in the nonlinear regime without going through the exponential gain regime at all.

2. clarity of presentation of the bunching scheme.

My understanding in the field of PBA is limited, and it was hard for me to follow the analytical derivation of the wake field beam bunching and acceleration process. The paper may be more comprehensible for experts in this field. Yet, since the readership community of Science Communications is diverse, it should be readable by people (at least people in the field of FEL) who are not experts in PBA. The paper needs to be revised to make it easy to read and comprehend. To help the authors understand the difficulty in reading the paper, I point out some points where I had difficulty to follow:

-In page 2 and Fig. 1(b) the authors talk about two cases "pre-modulated plasma" and "laser modulated plasma". The distinction is not clear, especially because it is stated that the pre-modulated plasma is generated by (two) laser beams modulation. The overlaid blue and red curves are confusing. Only when the reader arrives to line 384 he gets to understand that the blue line is a model and the red is calculated numerically for a specific model of laser pulses. But then, why isn't the blue curve uniform? Why are the curves overlaid? Did I correctly understand the intention?

-What are the axes and scales in Fig. 1(a)? Where is $z=0$?

-Lines 83-84: explain to the novice why injection is turned on and off at subluminal and superluminal phase velocity. Can this be explained in terms of trapping potentials?

-Line 87: how big is λ_w ?

-Line 89: mistake in the equation?

-Line 121: short laser pulses?

-Line 128: The equation is confusing, since it looks like some puzzling resonance condition. Only when I arrived at line 368 I understood that " π " is an arbitrary choice of example. I suggest the authors to use "3" instead of "pi" or clarify in line 128 that " π " is just an arbitrary choice of a number.

-line 172: define ξ as a function of z_i .

Supplementary Eq. 1: Define all the parameters in the equations.

Schnitzer, I., and A. Gover. "The prebunched free electron laser in various operating gain regimes." *Nuclear Instruments and Methods in Physics Research Section A: Accelerators, Spectrometers, Detectors and Associated Equipment* 237.1-2 (1985): 124-140.

Gover, A., and P. Sprangle. "A unified theory of magnetic bremsstrahlung, electrostatic bremsstrahlung, Compton-Raman scattering, and Cerenkov-Smith-Purcell free-electron lasers." *IEEE Journal of Quantum Electronics* 17.7 (1981): 1196-1215.

Reviewer #2 (Remarks to the Author):

The manuscript reports about the application of a density modulated downramp in a PWFA to produce electron beams pre-bunched at nm wavelengths, and of brightness high enough to lase in SASE mode, but now with high longitudinal coherence. In this way, previously reported density modulations at the 100 nm scale would be shortened by one to two orders of magnitude. Self-consistent PIC simulations of the electrons, generated in the ponderomotive potential determined in turn by the superposition of two counter-propagating lasers, are discussed, and results presented in terms of pre-bunched electron beam phase spaces.

The manuscript collects and profits of previously documented techniques, experimental as well as numerical achievements in PWFA, to propose a novel manipulation of the electron beam in the plasma. If successfully implemented, the scheme could open new roots towards the production of short wavelength x-ray FELs with high degree of longitudinal coherence. The manuscript is well-presented and scientifically sound.

Still, some doubts remain about the feasibility of some of the announced results, including electron beam manipulation up to the undulator and lasers synchronization, and the advantage of the overall scheme in the light of fully coherent XUV FELs already operating in, or targeting soon, the 1-4 nm range. Point-to-point comments and questions follow.

1. Abstract. It is certainly true that electron beams modulated at the wavelength of emission would enhance the longitudinal coherence of SASE FELs. This is already done at externally seeded and self-seeded FELs. The authors should instead highlight why the proposed scheme would bring some advantages, either physical or technological, or related to cost-effectiveness of the light source facility, with respect to seeded FELs in the same wavelength range. Please let me add that, while the scientific advancement illustrated by the manuscript is a clear and a valuable point per se, a discussion on why the proposed scheme should be evaluated versus present and coming-soon seeded FELs is missing (in the manuscript body).

1a. On the same subject, the advantage of the proposed scheme respect to existing self- and externally seeded FELs driven by RF accelerators remain unclear, once we consider a 2 GeV accelerator for the drive beam, plus a 1 GeV accelerator for the pre-modulated beam. A 3 GeV linac would be able to lase in HGHG or EEHG well below 1 nm with standard undulator technology. Still, microbunching instability could pose a limitation to the output performance. This same limitation, I think, would apply to the pre-modulated beam treated here. So, where is the real gain, if intended in terms of FEL user facilities?

2. The authors explain in Methods that the pre-bunched electron beam is accelerated to 1 GeV or so, and that this process is accomplished with the OSIRIS PIC code. It might be surprising the fact that the nm wavelength modulation in such low emittance, extremely high current beam, be not disrupted by longitudinal space charge. In other words, the authors claim the absence of any microbunching instability, which results one of the main show-stoppers to longitudinal coherence in x-rays. Could the authors comment and confirm that the PIC code has been taking care of local Coulomb interactions internal to the bunch during acceleration from 40 MeV to 1 GeV?

2a. Similar considerations bring the need of a straight layout from the plasma cell to the undulator, to avoid any ballistic and coherent synchrotron radiation-induced smearing of the fine modulation. Please confirm.

3. Figure 2e. The large current spike at the bunch edge (> 50 kA versus 17 kA of the bunched beam core) would most likely dominate any other SASE from the beam core. Could the author estimate the “signal-to-noise” ratio of such emissions (core vs. edge)? Or, propose any scheme for removal of the spike? (For example, beam scraping could be considered if necessary, but this would require the passage through a dispersive region, thus ballistic and collective smearing of the nm-bunching – see point 2a).

4. Demonstration of the suitability of the pre-bunched beam for lasing should be proven at least via numerical simulations of the FEL process. The authors are encouraged to proceed through this additional step, if they want to support their claim in the abstract as for unprecedented FEL brilliance level. This step would be relatively straightforward once the particle distribution in Fig.3c were imported into a 3-D FEL code like Genesis or Ginger, etc.

5. The feasibility of the proposed scheme might be limited by the capability of producing a 2 GeV beam with 34 kA peak current as for the driver. What kind of accelerator could do that, and which beam quality would be required (emittance, energy spread, mean energy and peak current jitter, pointing jitter)?

Line 134 and below: I suggest to inserting a Table summarizing the electron beam (driver, plasma-accelerated) and lasers parameters in practical units (e.g. International System).

Line 186: what is the timing/phase tolerance on the injection of the electron beam w.r.t. the driver, to minimize the projected energy spread? And what energy spread variations would this tolerance imply?

Line 134, 2nd column: what is the tolerance on the synchronization of the two counter-propagating lasers? As far as I know, state-of-the-art synchronization of two external EUV lasers from individual pumps stays at few fs level rms, shot-to-shot.

Line 89. The usage of k_w and λ_w is somehow confusing, as one would expect equality between the wavelength and the inverse wavenumber, if referring to the same quantity. Please clarify shortly by recalling the definition of the parameters involved.

Line 125-126: phrasing sounds wrong, please verify.

Line 146, 2nd column: the threshold on $k_{p0} \lambda_i$ should be put in absolute value, for consistency with Fig.3e.

We thank the referees for carefully reading and reviewing the paper, and for their constructive comments on how to enhance it. Please, find below point by point responses to the different questions raised, with a summary of the associated and substantial changes applied to the manuscript and the supplement. The referees' comments are shown in blue while our replies are black. The changes with respect to the previous versions of the main text and supplement have also been highlighted in red.

REVIEWER COMMENTS

Reviewer #1 (Remarks to the Author):

This manuscript describes an original scheme for bunching the density of a plasma wakefield accelerated electron beam at high harmonic frequencies up to the X-Ray regime. It is proposed that injecting such a beam into an undulator would start an FEL gain process without seed radiation injection or without relying on the wide band spectrum of electron beam shot-noise in SASE FEL. Such a coherent radiation emission process is essentially a bunched-beam coherent superradiance process in the high-gain regime. Pre-bunching of an electron beam can be generated with laser beams by the processes of HGHG and EEHG after the beam is accelerated to high energies. However, the proposed concept here is radically different, and is specific to the scheme of plasma wakefield acceleration, in which case the density modulation of the electrons is done by spatial density modulation of stationary plasma with two counter propagating laser beams. The bunched charge is then self-injected and accelerated by the plasma wake field of a short high energy electron beam pulse.

This concept looks like a promising way to utilize the relatively compact high acceleration gradient of PBA for operating an X-Ray FEL. The prebunching of the beam would enhance the FEL coherence and will expediate the FEL radiative power generation in a short undulator section (1 meter). The paper is in my mind of high value, and should be of interest to the scientific community of Science Communications readers. However, in my opinion it is not written in a clear and comprehensible way in its present form. Also, while the explanation of the bunching and plasma wakefield acceleration is detailed, the calculation of the gain and power of the pre-bunched beam FEL was done very briefly, possibly with not well fitting expressions that were derived earlier for a SASE FEL in another paper that is referred to. The FEL gain regime parameters are not given and the high power prediction is not well justified in the text or in a supplementary. I listed specific comments in an attachment, and I hope they will help the authors to improve the manuscript. I favor publication of this paper after substantial revisions.

Reviewer: Avraham Gover

Response: Dear Prof. Gover, thank you very much for your careful reading of our manuscript and your suggestions and constructive feedback. Below, we provide point by point responses to your comments.

I will present my comments in two parts: comments regarding the estimate of X-Ray power generation with the proposed beam bunching scheme and comments regarding the clarity of presentation of the bunching scheme.

1. comments regarding the estimate of X-Ray power generation with the proposed beam bunching scheme

The authors proposed a scheme of generating a periodically "pre-bunched electron beam at Angstrom scale by exciting a nonlinear wakefield across a periodically modulated plasma-density downramp/plasma cathode". They assert that upon injection of the modulated beam into a short undulator, high order harmonic components of the modulated beam current will excite coherent high power FEL radiation (for example 0.3 TW at 3.6 nm with a 1.09 GeV e-beam). This estimate is justified hastily by relying on the SASE FEL formulas of Xie [51]. The idea of prebunching the electron beam with high harmonic contents at X-Ray frequency is very interesting and may be important. While the main focus of the paper is the acceleration and bunching process, I expect that the estimate of the impressive high power predicted by the authors would be justified in more details and rigor, at least in the supplementary:

-Which formula in Xie's paper is used, the exponential linear gain formula? The saturation power formula?

Response: Eqs. 6 and 7 in Xie's paper were used to calculate the gain length and the saturation power, i.e., $\frac{L_{1d}}{L_g} = \frac{1}{1+\eta}$ and $P_{sat} \approx 1.6\rho \left(\frac{L_{1d}}{L_g}\right)^2 P_{beam}$, where ρ is the Pierce parameter or the FEL parameter and η is an intermediate variable which is defined in Xie's paper.

The spot size of the beam is set as a free parameter when using the fitting formula in Xie's paper, and we scan it to maximize the saturation power and minimize the gain length. Based on the concerns of the reviewer we instead now provide GENESIS 1.3 simulations where the prebunched beam is used.

-The high gain formula in Xie's paper refers specifically to SASE FEL, where the input power is the beam shot-noise (spontaneous emission). In the present case the input is coherent current pre-bunching. What is the analytical formula used? Short of simulation of the FEL problem (that is probably out of the scope of the present loaded paper) one can use an analytic solution of the cubic equation with initial conditions (in particular, initial condition of current density bunching). I can suggest the paper of [Schnitzer] with which I am familiar, but there are probably more recent references.

Response: The referee is right that the fitting formula in Xie's paper is derived for SASE. We use the fitting formula to give an order of magnitude estimation of the gain length and the saturation power for the prebunched case since the saturation power and gain length are similar for both cases. GENESIS simulations have now been added to show the FEL interaction for the prebunched case (see the following response). We have removed the fitting formulas since they were given as a guide for the reader but in the revised manuscript, we give results of GENESIS simulation instead.

-besides current modulation, the solution of the cubic equation can be given also in terms of velocity (energy) modulation. This may be relevant to the example in the paper where energy modulation is predicted (line 209). This energy modulation seems to be significant, since it can enhance the density bunching parameter from 0.05 to 0.24 with a short chicane (line 308). Since the wiggler is dispersive too, the energy modulation may play a role in the FEL dynamics in a meter long undulator.

Response: To demonstrate this bunched beam can produce high-power coherent radiation in a short undulator, we have conducted Genesis 1.3 simulations to model the FEL process. The basic design for matching the beam exiting the plasma to the undulator without significant emittance growth and debunching is shown in Fig. R1. Due to the ultra-strong focusing fields of the plasma wake, the injected beam has an extremely small spot size ($\sigma_r \approx 17 \text{ nm}$) and β -function ($\beta = \sqrt{2\gamma_b} \frac{c}{\omega_{p0}} \approx 80 \mu\text{m}$), where the beam energy is 1.09 GeV and the background plasma density is $n_{p0} = 1.97 \times 10^{19} \text{ cm}^{-3}$. Plasma matching sections and magnetic quadrupoles can be combined to transport the beam to the undulator with an enlarged β -function and conserved emittance (for example, see Ref. [3] for experimental demonstration). Adiabatic and non-adiabatic plasma matching sections have been studied extensively in the past decade (see Refs. [47-49]). For simplicity, here we use a plasma matching section solely with density profile as $n_p(z) = \frac{n_{p0}}{(1+z/l)^2}$ to enlarge the injected beam's β -function, where l is the characteristic length of the matching plasma (see Ref. [49] on how to choose the parameters).

Fig. R1 A conceptual plot of the plasma-based accelerator driven XFEL. See the text below for the actual distances and sizes of the various physical elements shown above.

We first boost the energy of the injected electrons for the case $g = 0.002$, $\lambda_L = 0.8 \mu\text{m}$ to 1.09 GeV by running OSIRIS simulation with ~ 3 million more cpu-hours. The phase space distribution of the injected electrons from the quasi-3D OSIRIS simulation is interpolated to six-dimensional phase space. After a 1.27 mm long plasma injection and acceleration section, the beam encounters the plasma density downramp matching section with $l \approx 47 \mu\text{m}$ and $L = 0.1 \text{ m}$, the beam has $\sigma_r \approx 1.4 \mu\text{m}$, $\beta \approx 0.56 \text{ m}$ and $\alpha \approx -1.3$. Here l is the distance over which the plasma density drops by a factor of 4 and L is the total length of the plasma. Then the beam drifts $L_d = 0.1 \text{ m}$ in free space to reach the entrance of the undulator and has $\sigma_r \approx 1.8 \mu\text{m}$, $\beta \approx 0.9 \text{ m}$, $\alpha \approx -1.8$. The plasma matching section and the drift are modelled by the corresponding transfer matrix. For the values of L and L_d used here the bunching factor is preserved. The 6D phase space distribution of the injected beam at the end of the drift is imported into GENESIS 1.3 with preserved current and energy modulation. The GENESIS 1.3 simulation result is shown in Fig. R2. The peak power of the 3.6 nm radiation grows exponentially and saturates around $z = 0.3 \text{ m}$ with $P \approx 234 \text{ GW}$. Note the external focusing magnets are absent and the natural focusing

force from the undulator can be neglected in such a short distance, the spot size of the beam grows by a factor of ~ 2 by the end of the simulation ($z = 0.6 \text{ m}$).

Fig. R2 The peak power of the radiation along the undulator. The fitted gain length is $L_{gain} \approx 4 \text{ cm}$ and the power at $z = 0.3 \text{ m}$ is $P \approx 234 \text{ GW}$. The parameters of the undulator: $\lambda_U = 1 \text{ cm}$, $K = 2.13$.

We have modified the Discussions section to reflect the above simulation result (see lines 361-371) and added a section in Methods to describe how the GENESIS simulation is done (see lines 498-520). A section has been added in the supplement with the above Figures R1 and R2 (see section 5 of the supplement).

-Since the beam parameters in the proposed scheme are a unique new case, it is desirable to check the operating parameters of the FEL dispersion equation for this case and verify the gain regime of the FEL (see [Gover and Sprangle] for example). Is the interaction in the cold beam limit? Is the exponential gain in the tenuous beam limit or there is a modification due to collective interaction? What is the gain e-folding length?

Response:

As you suggest, we start from your work in Ref. [Gover and Sprangle]. The definition of the cold beam limit is the normalized thermal spread parameter $\bar{\theta}_{th} \equiv \frac{2\pi v'_{th}}{\beta_{0z} v_{0z}} \frac{1}{\lambda_r} L_{sat} \ll 1$, where λ_r is the radiation wavelength and L_{sat} is the saturation length, and the meaning of other symbols can be found in [Gover and Sprangle]. The thermal spread $\bar{\theta}_{th}$ can be rewritten as $\bar{\theta}_{th} \approx 2\pi \frac{p_{zth}/m_e c}{\gamma_b^3 (1+K^2/2)^{-1}} \frac{1}{\lambda_r} L_{sat}$. If we substitute the parameters for the case studied in Fig. R2: 1.09 GeV mean beam energy, 0.4 MeV slice energy spread, 3.6 nm radiation wavelength and 0.3 m saturation length, we have $\bar{\theta}_{th} \approx 0.14$ which indicates the regime is in the cold limit.

The definition of the tenuous beam limit in Ref. [Gover and Sprangle] is $\frac{Q^{1/3}}{\theta_p} \gg 1$, where the gain parameter Q for a linearly polarized undulator is defined as $Q \equiv \left(\frac{K^2 [JJ]^2}{2} \frac{\pi A_e}{\lambda_U A_g} \right) \frac{\omega_p^2}{\gamma_b^3} \frac{c}{v_{0z}^3}$ and

the space charge parameter $\theta_p \equiv \sqrt{\frac{\omega_p^2}{\gamma_b^3(1+K^2/2)^{-1} v_{0z}}}$. If we substitute the following parameters:

$K = 2.13, \lambda_U = 1 \text{ cm}, E_b = 1.09 \text{ GeV}, I_b \approx 17 \text{ kA}, \sigma_r \approx 1.8 \text{ } \mu\text{m}$ and $\frac{A_e}{A_g} = 1$, we can get $\frac{Q^{1/3}}{\theta_p} \approx$

2.1 which indicates the interaction is close to the tenuous beam limit. Note ω_p in the above

expressions is calculated as $\omega_p^2 = \frac{I_b}{2\pi\sigma_r^2 ec} \frac{e^2}{m_e \epsilon}$.

The gain e-folding length is $L_{gain} \approx 4 \text{ cm}$ as shown in Fig. R2 and the corresponding FEL

parameter is $\rho \approx \frac{\lambda_U}{4\pi\sqrt{3}L_{gain}} \approx 0.01$. We have added text to the paper to point out we are in the cold/tenuous beam limit (see lines 370-371).

- The description of the contribution of reference [4] in the paper in lines 14-18 is not accurate. A more correct description would be in my words: Prebunching of the electron beam before injection into the FEL interaction region can produce coherent radiation emission at high harmonics of the bunching frequency, enhance the exponential gain of the FEL radiation power, and also can enable high power high radiative energy extraction efficiency by tapering-enhanced superradiance (TES) [4].

Response: We thank the referee for his suggestion for improving our text. We have changed the sentence as the referee suggested (see lines 14-20).

- Incidentally, I suggest the authors to examine the scheme of self-interaction TES of Ref. 4, where the bunched beam is trapped by its own superradiant radiation emission in a tapered undulator. Quite high trapping efficiency and high-power extraction may be attained in this case in the nonlinear regime without going through the exponential gain regime at all.

Response: We appreciate the referee's suggestion. Genesis simulation shows the interaction in a uniform undulator is in the exponential gain regime. We will study the interaction with a tapered undulator in the future.

2. clarity of presentation of the bunching scheme.

My understanding in the field of PBA is limited, and it was hard for me to follow the analytical derivation of the wake field beam bunching and acceleration process. The paper may be more comprehensible for experts in this field. Yet, since the readership community of Science Communications is diverse, it should be readable by people (at least people in the field of FEL) who are not experts in PBA. The paper needs to be revised to make it easy to read and comprehend. To help the authors understand the difficulty in reading the paper, I point out some points where I had difficulty to follow:

-In page 2 and Fig. 1(b) the authors talk about two cases "pre-modulated plasma" and "laser modulated plasma". The distinction is not clear, especially because it is stated that the pre-modulated plasma is generated by (two) laser beams modulation. The overlaid blue and red curves are confusing. Only when the reader arrives to line 384 he gets to understand that the blue line is a model and the red is calculated numerically for a specific model of laser pulses. But then, why isn't the blue curve uniform? Why are the curves overlaid? Did I correctly understand the intention?

Response: We have taken the reviewer's comments to heart and have rewritten parts of the paper. With respect to the reviewer's specific example, "pre-modulated plasma" refers to the case where both plasma electrons and ions are modulated. This can be achieved by many

methods, such as Ref. 40. The “laser modulated plasma” refers to the case where the plasma electrons are modulated by the ponderomotive force of the two laser pulses while the ions are barely modulated. These two cases represent two major categories of plasma density modulation, and we show that they are equivalent in the sense of modulating the phase velocity of a plasma wake in the supplement. We have modified the discussions on Page 3 (see lines 141-163) to make it clear.

why isn't the blue curve uniform? – The referee is right that the blue curve should be uniform. The non-uniformity of the blue line in our plot is because the resolution of the simulations is not high enough to catch the position where $E_z = 0$ exactly. Simulations with high resolution are now included in an updated plot (see Fig. R3 and the updated Fig. 1 in the manuscript). We can see the blue curve is more uniform now while the red curve is not as uniform as the blue one because the density modulation induced by two counter-propagating laser is more complicated than a sinusoidal profile (as discussed in the supplement).

Why are the curves overlaid? – This is because we carefully chose the parameters in the pre-modulated and laser-modulated cases to make sure the modulation density amplitude is similar in the two cases. See “Production of Fig. 1(b)” section in Methods for the detailed parameters.

-What are the axes and scales in Fig. 1(a)? Where is $z=0$?

Response: We have added axes to Fig. 1(a).

$z = 0$ is defined as the start of the plasma in this paper. We have added a symbol in Fig. 1(a) to point out where $z = 0$ is. Note Fig. 1(a) is a conceptual plot, and its elements are not to scale.

Fig. R3 The modulation of the phase velocity of position $E_z = 0$.

-Lines 83-84: explain to the novice why injection is turned on and off at subluminal and superluminal phase velocity. Can this be explained in terms of trapping potentials?

Response:

The reason that injection can be turned off and on can be qualitatively answered using trapping potentials. For a static wake there is a rigorous trapping condition, that arises from a constant of the motion, which for electrons is: $\gamma - v_\phi \frac{p_z}{m_e} - \frac{e}{m_e c^2} \psi = Const = 1$, where $\psi \equiv \phi - \frac{v_\phi}{c} A_z$ is the wake potential. The constant was evaluated for a plasma electron that starts at rest ahead of the drive. In order for a particle to be trapped its forward velocity v_z must be larger than v_ϕ . In a conventional accelerator a trapping condition is applied to externally injected particles where the constant has to be properly evaluated. In this case the injected electrons originate from within the plasma, and they are also part of the accelerating structure. The electrons within this structure form a narrow sheath that forms the boundary of the structure. So, to determine if some can be

trapped, it is also important to realize that the momentum of the electrons as they return to the axis at the rear of the first wavelength depends on the amplitude of the wake. To illustrate this, we show that trajectories of plasma electrons in a static wake. In Fig. R4 (reproduced from Fig. 2a in Ref. [25]), the grey lines show the trajectories of most plasma electrons while the colored lines show the electrons which have the highest possibility to be injected. These colored electrons move along the high-density wake sheath to the end of the wake. They obtain a large forward velocity v_z (close to the speed of light for an intense driver) and a small transverse velocity when they are at the end of the wake (shaded region in Fig. R4). When a particle is injected ($v_z = v_\phi$), thus the wake potential must be sufficiently negative at the rear of the wake such that $\gamma(v_z = v_\phi) - v_\phi \frac{p_z(v_z=v_\phi)}{m_e} = 1 + \frac{e}{m_e c^2} \psi$, which is usually called as “injection condition”. The left hand of the injection condition is a monotonically decreasing function of v_ϕ , which indicates that if v_ϕ is reduced then for a given ψ it is easier for injection to occur. Conversely for a given wake amplitude there is a threshold for v_ϕ for which self-injection occurs. The minimum value of ψ in a static wake is determined by the driver and the plasma density, which is $\frac{e}{m_e c^2} \psi_{min} \approx -0.98$ for the parameters studied in our work. Thus, injection can occur when $v_{\phi,th} \approx 0.9998c$ ($\gamma_{\phi,th} \approx 50$) (the transverse momentum of the particle is assumed to be 0). We can control v_ϕ by using a density downramp. For the modulated downramp where v_ϕ is no longer constant then the constant of the motion is no longer exact. However, if one still assumes it is approximately correct and we assume that v_ϕ is modulated by the density modulations then injection occurs when $v_\phi < v_{\phi,th}$ and is not possible if $v_\phi > v_{\phi,th}$. Since $v_{\phi,th}$ is very close to the speed of light, we approximately say the injection is turned on and off at subluminal and superluminal phase velocities. In Fig. 1(b) of the main body, the modulation amplitude of the phase velocity of the position $E_z = 0$ (roughly the middle of the wake) is $0.005c$. We can infer the modulation amplitude of v_ϕ of the end of the wake is roughly twice of $0.005c$, i.e., $0.01c$. Thus, its phase velocity oscillates between $v_\phi \approx 0.99c$ and $v_\phi \approx 1.01c$ and the injection is turned on and off.

We have modified the text to include this discussion (see lines 95-102).

Fig. R4 The trajectories of the plasma electrons in a static nonlinear plasma wake. This figure is attribution to the X. L. Xu, F. Li, W. AN, T. N. Dalichaouch, P. Yu, W. Lu, C. Joshi, W. B. Mori and

High quality electron bunch generation using a longitudinal density-tailored plasma-based accelerator in the three-dimensional blowout regime, *Physical Review Accelerators and Beams* 20, 111303 (2017), and 10.1103/PhysRevAccelBeams.20.111303.

-Line 87: how big is λ_w ?

Response: λ_w is the length of the first plasma wave bucket, which depends on the driver and the plasma density. It is $9k_{p0}^{-1}$ for the beam driver used in this paper, which is $10.8 \mu\text{m}$ if $n_{p0} = 1.97 \times 10^{19} \text{cm}^{-3}$.

We have modified this sentence to make it clear (see line 114).

-Line 89: mistake in the equation?

Response: We have fixed it (see line 117).

-Line 121: short laser pulses?

Response: We have fixed it (see line 153).

-Line 128: The equation is confusing, since it looks like some puzzling resonance condition. Only when I arrived at line 368 I understood that “ π ” is an arbitrary choice of example. I suggest the authors to use “3” instead of “pi” or clarify in line 128 that “ π ” is just an arbitrary choice of a number.

Response: We have added a sentence (see line 163) to make it clear.

-line 172: define ξ as a function of z_i .

Response: We have added one sentence (see lines 200-202) to define it and modified line 206 to make it clear.

Supplementary Eq. 1: Define all the parameters in the equations.

Response: We have added definitions of all the parameters after Eq. 1 (see page 2 of the supplement).

Schnitzer, I., and A. Gover. "The prebunched free electron laser in various operating gain regimes." *Nuclear Instruments and Methods in Physics Research Section A: Accelerators, Spectrometers, Detectors and Associated Equipment* 237.1-2 (1985): 124-140.
Gover, A., and P. Sprangle. "A unified theory of magnetic bremsstrahlung, electrostatic bremsstrahlung, Compton-Raman scattering, and Cerenkov-Smith-Purcell free-electron lasers." *IEEE Journal of Quantum Electronics* 17.7 (1981): 1196-1215.

Reviewer #2 (Remarks to the Author):

The manuscript reports about the application of a density modulated downramp in a PWFA to produce electron beams pre-bunched at nm wavelengths, and of brightness high enough to lase in SASE mode, but now with high longitudinal coherence. In this way, previously reported density modulations at the 100 nm scale would be shortened by one to two orders of magnitude. Self-consistent PIC simulations of the electrons, generated in the ponderomotive potential determined

in turn by the superposition of two counter-propagating lasers, are discussed, and results presented in terms of pre-bunched electron beam phase spaces.

The manuscript collects and profits of previously documented techniques, experimental as well as numerical achievements in PWFA, to propose a novel manipulation of the electron beam in the plasma. If successfully implemented, the scheme could open new roots towards the production of short wavelength x-ray FELs with high degree of longitudinal coherence. The manuscript is well-presented and scientifically sound.

Still, some doubts remain about the feasibility of some of the announced results, including electron beam manipulation up to the undulator and lasers synchronization, and the advantage of the overall scheme in the light of fully coherent XUV FELs already operating in, or targeting soon, the 1-4 nm range. Point-to-point comments and questions follow.

1. Abstract. It is certainly true that electron beams modulated at the wavelength of emission would enhance the longitudinal coherence of SASE FELs. This is already done at externally seeded and self-seeded FELs. The authors should instead highlight why the proposed scheme would bring some advantages, either physical or technological, or related to cost-effectiveness of the light source facility, with respect to seeded FELs in the same wavelength range. Please let me add that, while the scientific advancement illustrated by the manuscript is a clear and a valuable point per se', a discussion on why the proposed scheme should be evaluated versus present and coming-soon seeded FELs is missing (in the manuscript body).

Response: We have changed the abstract as the referee suggested and added a discussion in the body of the manuscript (see line 55-67, 74-77).

1a. On the same subject, the advantage of the proposed scheme respect to existing self- and externally seeded FELs driven by RF accelerators remain unclear, once we consider a 2 GeV accelerator for the drive beam, plus a 1 GeV accelerator for the pre-modulated beam. A 3 GeV linac would be able to lase in HGHG or EEHG well below 1 nm with standard undulator technology. Still, microbunching instability could pose a limitation to the output performance. This same limitation, I think, would apply to the pre-modulated beam treated here. So, where is the real gain, if intended in terms of FEL user facilities?

Response:

Here the referee is asking about the microbunching instability which has shown to reduce the performance of LCLS X-FEL unless laser heating is used to condition the beam and kill the instability. There are far fewer magnetic optics between the generation of the bunched beam and its eventual injection into the undulator. We have done a self-consistent simulation of beam matching to the undulator and have not seen any gross reduction of the beam emittance, charge, or bunching factor due to the microbunching instability. One aspect of the microbunching instability, the beam plasma oscillation, is discussed in detail in the response to Question 2.

The advantage is that plasma-based schemes produce beams with brightnesses 3~4 orders of magnitude higher than current technology and the accelerator can also potentially be 2~3 orders

of magnitude shorter. The higher brightnesses could lead to X-ray powers several orders of magnitude higher than existing technology. We next give specifics for each of these points.

(i) As we explain below (Response to comment 5), a ~ 700 MeV beam driver is enough to produce the 1 GeV pre-modulated beam. Furthermore, the proposed scheme can also be realized by using a laser pulse as the driver. So, we have three options for the driver: i) ~ 700 MeV beam driver produced from RF-based accelerators, ii) a ~ 700 MeV beam driver produced from a laser-driven plasma wakefield accelerator; and iii) a laser pulse driver. The proposed scheme with the first option can be viewed as a brightness booster since the brightness of the injected beam from plasma is ~ 3 orders of magnitude higher than the beam driver from RF accelerators. In addition, the beam quality of the driver can be relaxed so that the rf accelerator required to produce the high brightness beam can be less complex than those currently required to drive an XFEL. The proposed scheme with the second and third options makes the GeV-class pre-bunched beam generation much more compact than conventional schemes because the millimeter~centimeter acceleration distance and the avoidance of chicane and modulators.

(ii) Due to the ultra-high brightness and initial bunching factor, the injected pre-bunched beam can produce nanometer wavelength radiation with hundreds of GW in a meter-long undulator. For example, we did Genesis 1.3 simulations to show 3.6 nm coherent radiation with 0.2 TW peak power in a 0.3-meter undulator. This power is 2 orders of magnitude higher than other harmonic seeding techniques. A short undulator further decreases the size of the proposed scheme and reduces its cost.

(iii) We also note that if the plasma accelerator section can be tuned then even higher output powers could potentially be produced. In principle, the energy of the injected beams when they achieve a flat longitudinal phase space can be increased by using shallower ramps ($g < 0.002$) and their peak current can be increased by using more intense drivers. These improvements can produce more radiation power in a resonant undulator, which may reach TW-level.

We are not claiming that the proposed a scheme will replace the RF accelerator based XFELs. We think the proposed novel scheme can be a favorable complement to current existing XFEL facilities or may be used as a brightness booster for existing or future XFELs.

2. The authors explain in Methods that the pre-bunched electron beam is accelerated to 1 GeV or so, and that this process is accomplished with the OSIRIS PIC code. It might be surprising the fact that the nm wavelength modulation in such low emittance, extremely high current beam, be not disrupted by longitudinal space charge. In other words, the authors claim the absence of any microbunching instability, which results one of the main show-stoppers to longitudinal coherence in x-rays. Could the authors comment and confirm that the PIC code has been taking care of local Coulomb interactions internal to the bunch during acceleration from 40 MeV to 1 GeV?

Response:

The self-space-charge interaction is fully included in the PIC code. The energy modulation is essentially a result of the space-charge interactions between the injection electrons. In Fig. R4, we focus on a small part of the injected beam to show how the energy modulation is established. When the electrons are injected, there is a current modulation but negligible energy modulation

(blue line in Fig. R4). As the electrons are accelerated inside the plasma wake, their space-charge force establishes an energy modulation gradually. Since the plasma oscillation frequency of the injected beam depends on its energy as $\omega_{p,b} \propto \gamma^{-3/2}$, the energy modulation is mainly established when the beam energy is low and then remains frozen as it continues to be accelerated.

Due to the short acceleration distance (~ 1000 plasma skin depth, which is ~ 1 mm when $n_{p0} = 1.97 \times 10^{19} \text{ cm}^{-3}$) and the huge acceleration gradient, the energy modulation does not degrade the current modulation significantly. The beam with 1.09 GeV energy is still characterized by a clean current modulation.

Fig. R4 The energy modulation of the beam at different acceleration distance. The linear energy chirp is removed for comparison.

2a. Similar considerations bring the need of a straight layout from the plasma cell to the undulator, to avoid any ballistic and coherent synchrotron radiation-induced smearing of the fine modulation. Please confirm.

Response:

To get the beam out of the plasma and into an undulator requires “matching” sections at the end of the plasma. The basic design is shown in Fig. R1 where only plasma matching section and a drift space are used in a linear design.

3. Figure 2e. The large current spike at the bunch edge (> 50 kA versus 17 kA of the bunched beam core) would most likely dominate any other SASE from the beam core. Could the author estimate the “signal-to-noise” ratio of such emissions (core vs. edge)? Or, propose any scheme for removal of the spike? (For example, beam scraping could be considered if necessary, but this would require the passage through a dispersive region, thus ballistic and collective smearing of the nm-bunching – see point 2a).

Response:

(1) Signal to noise: As shown in Fig. R5, the large current beam head has a larger emittance and a larger energy spread than the beam core: $\sim 8 I_A$ v.s. $\sim I_A$, $\sim 0.06 k_{p0}^{-1}$ v.s. $\sim 0.006 k_{p0}^{-1}$ and ~ 16 MeV v.s. ~ 0.4 MeV. Thus, the 6D brightness of the beam head is lower by a factor of ~ 500 than that of the beam core. We expect the large current head generates much lower radiation compared with the pre-bunched beam core in a short undulator. The GENESIS simulation shows a power contrast as 234 GW (beam core) v.s. 8 GW (beam head) at $z = 0.3$ m. Note the above GENESIS simulation only models the region with a transverse size of 80 microns in each direction, therefore particles that make greater excursions are removed automatically.

(2) Removal of the spike: The current spike at the head of the beam is caused by a violent injection process around the start of the ramp (see Ref. 25). It can be eliminated when a small wake is excited or perhaps when a smoother density ramp is used. In Fig. R6, we can see the large current beam head disappears when a driver with low peak current (case 2) or large spot size is used (case 3).

We thank the referee for pointing out these issues and have added the above discussion and the Fig. R5 in the supplement section (see section 5.2 of the supplement).

Fig. R5 The current, emittance and energy spread of the injected beam in Fig. 2 at $145\omega_{p0}^{-1}$.

4. Demonstration of the suitability of the pre-bunched beam for lasing should be proven at least via numerical simulations of the FEL process. The authors are encouraged to proceed through this additional step, if they want to support their claim in the abstract as for unprecedented FEL brilliance level. This step would be relatively straightforward once the particle distribution in Fig.3c were imported into a 3-D FEL code like Genesis or Ginger, etc.

Response: See our response to referee 1 who had a similar suggestion. We have imported the data into Genesis 1.3 to model the FEL process. Please see the discussion of Figs. R1 and R2 (pages 3 and 4 of the response letter).

5. The feasibility of the proposed scheme might be limited by the capability of producing a 2 GeV beam with 34 kA peak current as for the driver. What kind of accelerator could do that, and which beam quality would be required (emittance, energy spread, mean energy and peak current jitter, pointing jitter)?

Response:

(i) What type of accelerator: These beams could be produced at FACET-II facility at SLAC National Accelerator laboratory and current XFEL facilities or at more than a dozen of 100 TW-PW class laser facilities around the world. We discuss both next. The role of the beam driver in the proposed scheme is to produce a fully nonlinear plasma wake. The major requirements of the beam driver are its peak current and three-dimensional sizes which needs to be on the order of or smaller than the plasma skin depth to excite a plasma wake effectively. A beam with $I_{pk} = 34 \text{ kA}$, $k_{p0}\sigma_z = 0.7$, $k_{p0}\sigma_r = 0.5$ is used in the paper, where $k_{p0}^{-1} \approx 1.2 \mu\text{m}$ if a background plasma density $n_{p0} = 1.97 \times 10^{19} \text{ cm}^{-3}$ is assumed. A driver with lower current or non-ideal three-dimensional sizes excites a smaller wake which would generate a bunched beam with larger harmonic number h and lower peak current (see case 2 and case 3 in Fig. R6 and the average current is $\sim 0.75I_A$ for case 3 and $\sim 0.3I_A$ for case 4).

The proposed scheme has large tolerance for the energy spread, the variations of the mean energy and the emittance of the driver since the plasma wake excitation is not sensitive to these parameters (see case 4 in Fig. R6). However, due to the limited transformer ratio (the ratio between the output beam energy change and drive beam energy change) which is ~ 1.6 for the parameters used in the paper, there is a minimum requirement of the energy of the driver beam, $\sim 700 \text{ MeV}$, if we want to accelerate the injected beam to 1 GeV.

Electron beams produced in laser-driven plasma wakefield accelerators are also suitable to be used as drivers in our proposed scheme. The scaling law tells an 800 nm laser pulse with $\sim 100 \text{ TW}$ power can produce a GeV-class electron beam in a plasma with $2 \times 10^{18} \text{ cm}^{-3}$ density [W. Lu, et al., Phys. Rev. ST Accel. Beams 10, 061301 (2007)]. Such laser pulses are available in many laser facilities. And the technology of using the self-injected beam from a laser plasma accelerator to excite a beam-driven plasma wake has been demonstrated experimentally [T. Kurz, et al., Nat. Commun. 12, 2895 (2021)].

Furthermore, the FACET-II facility has an experimental program for generating electron beams with 10~100 kA peak current and $\sim \text{fs}$ bunch duration [V. Yakimenko, et al., Phys. Rev. Accel. Beams., 22, 101301 (2019)]. Although the spot size of these beams is larger than we desire, they can be focused down by properly designing plasma upramp [C. Joshi, et al., Plasma Phys. Control. Fusion 60, 034001 (2018)]. Current XFEL facilities can produce ultra-short beams with several $\sim 10 \text{ kA}$ peak current which can also drive a density downramp injection [J. Grebenyuk, et al., Nucl. Instrum. Methods Phys. Res., Sect. A 740, 246 (2014); X. Xu et al., Physical Review Accelerators and Beams 20, 111303 (2017)].

Fig. R6 Comparison of the injected bunched beams when different beam drivers are used. (a) the current profile; (b) the bunching factor. To save the computational cost, we use a pre-modulated plasma with $\delta \bar{n} = 10^{-3} n_{p0}$, $g = 3 \times 10^{-3}$ and $k_{p0} \lambda_m = 1$. Case1: reference case, $E_d = 2 \text{ GeV}$, $\sigma_{E_d} = 0$, $\epsilon_{N,d} = 0$, $I_{pk} = 34 \text{ kA}$, $k_{p0} \sigma_z = 0.7$, $k_{p0} \sigma_r = 0.5$; case 2: low current driver case, $E_d = 2 \text{ GeV}$, $\sigma_{E_d} = 0$, $\epsilon_{N,d} = 0$, $I_{pk} = 17 \text{ kA}$, $k_{p0} \sigma_z = 0.7$, $k_{p0} \sigma_r = 0.5$; case 3: wide driver case, $E_d = 2 \text{ GeV}$, $\sigma_{E_d} = 0$, $\epsilon_{N,d} = 0$, $I_{pk} = 34 \text{ kA}$, $k_{p0} \sigma_z = 0.7$, $k_{p0} \sigma_r = 1.25$; case 4: $E_d = 0.5 \text{ GeV}$, $\sigma_{E_d} = 10 \text{ MeV}$, $\epsilon_{N,d} = 1 \mu\text{m}$, $I_{pk} = 34 \text{ kA}$, $k_{p0} \sigma_z = 0.7$, $k_{p0} \sigma_r = 0.5$.

High-quality pre-bunched electron beams might also be generated by using a laser driver to create the wake. Here, we show an example in Fig. R7. A linearly polarized 800 nm laser pulse with peak normalized vector potential $a_0 = 4$ propagates through the pre-modulated plasma downramp and injects a pre-bunched electron beam as shown in Fig. R7. The laser driver is not as rigid as the beam driver and evolves when it propagates in the plasma downramp. The evolution of the laser driver leads to an additional expansion of the wake during the downramp region which is equivalent to a ramp with larger gradient g . Thus, the harmonic number in this laser-driven case is $h \approx 77$, which is smaller than $h \approx 150$ in the beam-driven case. This is a preliminary simulation and thus it is far from optimized. The publication of this work will spur further research into how to optimize the generation of high harmonic numbers.

Using a particle beam driver (generated from a conventional accelerator or from an LFWA) or a laser driver have their own strong points and weaknesses: the tightly focused beam driver generated from laser-driven plasma accelerators can work in a plasma with higher density ($\gtrsim 10^{20} \text{ cm}^{-3}$) to produce pre-bunched beams with shorter bunching wavelength and smaller emittance while the scheme with a laser driver will lead to simpler and more compact designs.

We have added a paragraph in the manuscript to discuss the considerations on drivers (see lines 372-393). Details and Figs. R6, R7 have been added to the supplement (see section 6 of the supplement).

Fig. R7 The current profile (a) and the bunching factor (b) of the injected beam in a laser-driven plasma wake. Parameters: the laser has a spot size $w_0 = 7.6 \mu\text{m}$, and a duration $\tau_{FWHM} = 28.4 \text{ fs}$; a pre-modulated plasma downramp with $\delta\bar{n} = 10^{-3}n_{p0}$, $g = 3 \times 10^{-3}$ and $k_{p0}\lambda_m = 1$ is used and $n_{p0} = 7.73 \times 10^{18} \text{ cm}^{-3}$.

(ii) Jitter: Pointing jitter is a critical issue for all schemes using beams from plasma to drive a XFEL. Compared with other schemes, the injected beam here is characterized by a very high brightness and an initial bunching factor which can saturate the FEL process very quickly (~ 0.3 meter). Thus, the proposed scheme is less susceptible to the transverse position and angle jitters. We give a rough estimation of the tolerance.

In a typical high gain XFEL, transverse position and angle offsets result in an oscillation of the centroid of the electron beam about the undulator axis due to the focusing elements along the undulator, which can significantly degrade the performance of the machines [T. Tanaka et al., Nucl. Instrum. Methods Phys. Res., Sect. A 528, 172 (2004); P. Baxevanis, Z. Huang, and G. Stupakov, Phys. Rev. Accel. Beams 20, 040703 (2017)]. As a contrast, external focusing magnetic is absent in our proposed scheme and the natural focusing force from the undulator can be neglected in such a short distance, thus a transverse angle offset θ_{jitter} just results in a slant trajectory of the beam centroid. And its main consequence is a decrease of the undulator wavelength as $\lambda_U \cos \theta_{jitter}$. A crude estimation of the angle tolerance is that the relative shift of the undulator wavelength is much less than the FEL parameter ρ , i.e., $\frac{\Delta\lambda_U}{\lambda_U} \ll \rho$, which indicates $\theta_{jitter} \ll 0.14 \text{ rad}$ for $\rho \approx 0.01$.

Another consideration of the transverse jitters is that based on the variation of magnetic field of the undulator which is $B = B_0(1 + k_U^2 x^2) \cos k_U z$. The requirement is the relative variation of the magnetic field is much less than the FEL parameter, i.e., $k_U^2(x_{jitter} + \theta_{jitter}L_U)^2 \ll \rho$. Substitute the parameters, it gives $x_{jitter} + \theta_{jitter}L_U \ll 160 \mu\text{m}$. Recall the saturation length is 0.3 m, thus the requirements of the jitters are $x_{jitter} \ll \sim 100 \mu\text{m}$, $\theta_{jitter} \ll 0.5 \text{ mrad}$. These can be achieved in current ultra-short high-power laser facilities or electron beam accelerators.

Line 134 and below: I suggest to inserting a Table summarizing the electron beam (driver, plasma-accelerated) and lasers parameters in practical units (e.g. International System).

Response: We have added a Table in the supplemental section (see section 4 in the supplement).

Line 186: what is the timing/phase tolerance on the injection of the electron beam w.r.t. the driver, to minimize the projected energy spread? And what energy spread variations would this tolerance imply?

Response: The electrons are self-injected into wake, thus their positions are locked with the driver automatically.

Line 134, 2nd column: what is the tolerance on the synchronization of the two counter-propagating lasers? As far as I know, state-of-the-art synchronization of two external EUV lasers from individual pumps stays at few fs level rms, shot-to-shot.

Response:

The two counter propagating lasers can be of ps duration while the drive beam or laser pulse is of typically 30-50 fs duration. Therefore state-of-the art synchronization techniques are sufficient to meet the timing tolerance requirements.

Line 89. The usage of k_w and λ_w is somehow confusing, as one would expect equality between the wavelength and the inverse wavenumber, if referring to the same quantity. Please clarify shortly by recalling the definition of the parameters involved.

Response: Thank you for pointing this out. ' k_w ' is a typo which should be ' k_m ' which is the plasma density modulation wavenumber. We have corrected it (see line 117).

Line 125-126: phrasing sounds wrong, please verify.

Response: We have changed it from "The modulation of the phase velocity in both cases: a pre-modulated plasma and a laser-modulated plasma are shown in Fig. 1(b)." to "The modulation of the phase velocity from a pre-modulated plasma and a laser-modulated plasma are shown in Fig. 1(b)." See lines 157-159.

Line 146, 2nd column: the threshold on k_{p0} should be put in absolute value, for consistency with Fig.3e.

Response: We have changed it (see line 236).

REVIEWERS' COMMENTS

Reviewer #1 (Remarks to the Author):

The paper presentation is now clearer and I wholeheartedly recommend publication of the paper. Two comments:

-I think that the inequality sign in line 117 is in the wrong direction (or I do not understand the related sentence.

-The article suggests achievable high power in a X-Ray FEL. In the revised version this is supported by GENESIS simulation instead of the analytic expressions and is much more convincing. I therefore consider Fig. R2 to be an important demonstration of the claim of the paper, and it should be better placed in the main text rather than the supplementary.

Reviewer #2 (Remarks to the Author):

The authors addressed all the criticisms, they answered point-to-point and provided results solid enough to make the modified manuscript scientifically sound. The inclusion of FEL simulations is really appreciated, and I think it enriches the manuscript considerably w.r.t. its original version.

As a result, the study is specific but also comprehensive enough to illustrate a novel and promising approach to pre-modulated beam drivers for short wavelength, coherent FELs.

I am in favour of its publication in its present form.

REVIEWERS' COMMENTS

Reviewer #1 (Remarks to the Author):

The paper presentation is now clearer and I wholeheartedly recommend publication of the paper.

Response: We thank the referee for carefully reading of our paper and providing useful comments.

Two comments:

-I think that the inequality sign in line 117 is in the wrong direction (or I do not understand the related sentence.

Response: Thank you for pointing this typo out. We have fixed it.

-The article suggests achievable high power in a X-Ray FEL. In the revised version this is supported by GENESIS simulation instead of the analytic expressions and is much more convincing. I therefore consider Fig. R2 to be an important demonstration of the claim of the paper, and it should be better placed in the main text rather than the supplementary.

Response: We have moved Fig. R2 to the main text following the referee's suggestion.

Reviewer #2 (Remarks to the Author):

The authors addressed all the criticisms, they answered point-to-point and provided results solid enough to make the modified manuscript scientifically sound. The inclusion of FEL simulations is really appreciated, and I think it enriches the manuscript considerably w.r.t. its original version.

As a result, the study is specific but also comprehensive enough to illustrate a novel and promising approach to pre-modulated beam drivers for short wavelength, coherent FELs. I am in favour of its publication in its present form.

Response: We thank the referee for valuable feedback, and are pleased that we were able to address all concerns.